# TAKING THE GP OUT OF THE LOOP

## ABSTRACT

Bayesian optimization (BO) has traditionally solved black-box problems where evaluation is expensive and, therefore, observations are few. Recently, however, there has been growing interest in applying BO to problems where evaluation is cheaper and observations are more plentiful. Scaling BO to many observations, $N$, is impeded by the $\mathcal{O}(N^3)$ cost of a naïve query (or $\mathcal{O}(N^2)$ in optimized implementations) of the Gaussian process (GP) surrogate. Many methods improve scaling at acquisition time, but hyperparameter fitting still scales poorly. Because a GP is refit at every iteration of BO, fitting remains the bottleneck. We propose Epistemic Nearest Neighbors (ENN), a lightweight alternative to GPs that estimates function values and epistemic uncertainty from $K$-nearest-neighbor observations. ENN has $\mathcal{O}(N)$ acquisition cost and, crucially, omits hyperparameter fitting, making ENN-based BO also $\mathcal{O}(N)$. Because ENN omits hyperparameter fitting, its uncertainty scale is arbitrary, making it incompatible with standard acquisition methods. We resolve this by applying a non-dominated sort (NDS) to candidate points, treating predicted values ($\mu$) and uncertainties ($\sigma$) as two independent metrics. Our method, TuRBO-ENN, replaces the GP surrogate in TuRBO with ENN and its Thompson-sampling acquisition with this NDS-based alternative. We show empirically that TuRBO-ENN reduces proposal generation time by one to two orders of magnitude compared to TuRBO and scales to thousands of observations.

## 1 INTRODUCTION

Bayesian optimization (BO) is commonly used in settings where evaluations are expensive, such as A/B testing (days to weeks) Quin et al. (2023); Sweet (2023), materials experiments (roughly 1 day) Kotthoff et al. (2021). It has also been applied to simulation optimization problems in engineering, logistics, medicine, and other domains Amaran et al. (2017). More recently, BO has been used in settings where evaluations are fast and can be run in parallel—for example, large-scale simulations in engineering design. In such cases, thousands of evaluations may be generated during a single optimization process Daulton et al. (2021).

BO methods typically scale poorly with the number of observations, $N$, because proposals are generated by fitting and querying a Gaussian process (GP) surrogate. Modern, optimized implementations require $O(N^2)$ time per query. We refer to this setting as *Bayesian optimization with many observations (BOMO)*, and present a method that reduces the proposal-time scaling to $O(N)$.

It is important to distinguish between BOMO and BO with many design parameters – high-dimensional Bayesian optimization (HDBO). Generally, we expect to need more observations to optimize more parameters since there are simply more possible designs to evaluate. This expectation is codified, for example, in Ax's (Meta (2023)) prescription to collect $2 \times D$ observations before fitting a surrogate (where $D$ is the number of design parameters, or *dimensions*). However, the number of observations necessary to locate a good design depends on more than just $D$. For example, Wang et al. (2016) optimizes a one-billion-parameter function with only 500 observations, while Daulton et al. (2021) takes 1500 observations to optimize a simulator with only 12 parameters.

This work focuses on BOMO. Specifically, we ask: **Can we make a SOTA algorithm significantly faster on BOMO problems while producing comparable-quality solutions?** We are concerned mainly with scaling (with $N$) but we also report on wall time. Our approach is to strategically simplify, then compare solution quality, scaling, and running time. One could think of this paper as an ablation of the state-of-the-art in BO.

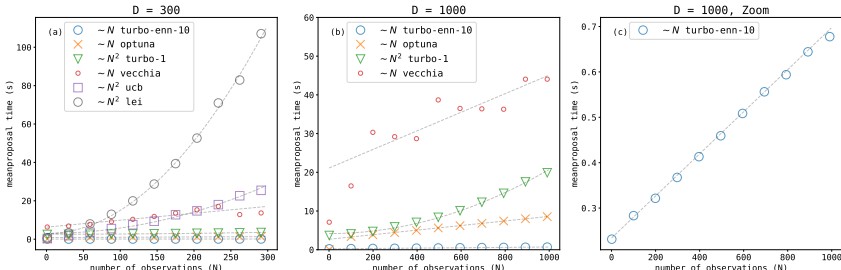

Figure 1: Mean proposal time (in seconds) versus number of observations ($N$) for several Bayesian optimization methods. Subfigures show results for (a) $D = 300$, (b) $D = 1000$, and (c) a zoomed-in view of (b), averaged over many optimization runs (see Section 5). GP-based methods (`lei`, `ucb`, `turbo-1`) scale approximately as $O(N^2)$, while `optuna`, which uses a Parzen estimator, `vecchia`, and our method (`turbo-enn-10`) scale linearly in $N$. Results are averaged over 51 functions × 10 BO runs/function = 510 runs for each optimization mathod.

We propose a method consisting of two components: (i) a $K$-nearest neighbors surrogate, Epistemic Nearest Neighbors (ENN), which estimates function values and (uncalibrated, Section 4) epistemic uncertainty, and (ii) an acquisition method compatible with uncalibrated uncertainty estimates. We integrate this approach into TuRBO Eriksson et al. (2019) by replacing its GP surrogate and Thompson sampling acquisition method with ENN and our acquisition method. Additionally, as many simulation optimization problems are effectively noise-free Santner et al. (2019, Chapter 1, Section 2), and, for clarity of exposition, we restrict attention to the noise-free case in this paper. Extensions to noisy settings are a natural direction for future work.

## 2 BACKGROUND

### 2.1 BAYESIAN OPTIMIZATION

A Bayesian optimizer proposes a design, $x \in= [0, 1]^D$, given some observations, $\mathcal{D} = \{(x_m, y_m)\}_{m=1}^N$, where $y_m = f(x_m)$. A typical BO method consists of two components: a surrogate and an acquisition method. A surrogate is a model of $f(x)$ mapping a design, $x$, to both an estimate of $f(x)$, $\mu(x)$, and a measure of uncertainty in that estimate, $\sigma(x)$. An acquisition method determines the proposal, $x_p = \arg\max_x \alpha(\mu(x), \sigma(x))$, where the $\arg\max$ is found by numerical optimization (e.g., via BFGS Meta (2024a)) or by evaluating $\alpha(\cdot, \cdot)$ over a set of $x$ samples, for example, uniform in $[0, 1]^D$ or following any number of sampling schemes Kandasamy et al. (2017); Eriksson et al. (2019); Rashidi et al. (2024a).

### 2.1.1 SURROGATE

The usual BO surrogate is a Gaussian process Rasmussen & Williams (2006). Given observations $\mathcal{D} = \{(x_m, y_m)\}_{m=1}^N$, the GP posterior at a new point $x$ has mean and variance

$$
\begin{aligned}
\mu(x) &= K(x_m, x)^\top K(x_m, x_m)^{-1} y_m \\
\sigma^2(x) &= 1 - K(x_m, x)^\top K(x_m, x_m)^{-1} K(x_m, x).
\end{aligned}
\tag{1}
$$

The $N \times N$ kernel matrix, $K(x_m, x_m)$, has as its elements $K(\cdot, \cdot)_{ij} = k(x_{m,i}, x_{m,j})$, where $k(\cdot, \cdot)$ is a kernel function, often a squared exponential $k(x_{m,i}, x_{m,j}) = e^{-\|x_{m,i} - x_{m,j}\|^2/2\lambda}$, although others are common, too Rasmussen & Williams (2006). Similarly, the kernel vector, $K(x_m, x)_i = k(x_{m,i}, x)$. The kernel matrix is the pairwise covariance between all observations, and the kernel vector is the covariance between the query point, $x$, and the observations.

The $N \times N$ kernel matrix is a source of the GP's $O(N^2)$ scaling with number of observations as it takes $N(N-1)/2$ evaluations of $k(\cdot, \cdot)$ to construct it. A straightforward calculation of $K(x_m, x_m)^{-1}$ would worsen the scaling to $O(N^3)$, but an efficient conjugate-gradient algorithm reduces this, also, to $O(N^2)$ Gardner et al. (2018).

The hyperparameter, $\lambda$, the kernel length-scale, is typically tuned to maximize the marginal log-likelihood of the observations, $\mathcal{D}$, by a numerical optimizer such as SGD Eriksson et al. (2019) or BFGS Meta (2024a).

### 2.1.2 Acquisition method

There are many acquisition methods in the literature. Three common ones are:

**Upper Confidence Bound (UCB)** $x_p = \arg\max_x [\mu(x) + \beta\sigma(x)]$, where $\beta$ is a constant. The first term encourages exploitation of $\mathcal{D}$, i.e. biasing $x_p$ towards a design that is expected to work well, while the second term encourages exploration of the design space so as to collect new observations that will improve future surrogates.

**Expected Improvement (EI)** $x_p = \arg\max_x \mathbb{E}[\max\{0, y(x) - y_*\}]$, where $y_* = \max y_m$, and the expectation is taken over $y(x) \sim \mathcal{N}(\mu(x), \sigma^2(x))$.

**Thompson Sampling (TS)]** $x_p = \arg\max_x y(x)$, where $y(x)$ is a joint sample from the GP at a set of $x$ values. (A joint sample modifies equations equation 1 to account for the covariance between each $x$ that is being sampled.)

All three methods rely on the GP's uncertainty being calibrated. Calibration, i.e. hyperparameter tuning, requires multiple queries of the GP, each of which takes $O(N^2)$ time.

We next introduce a surrogate and companion acquisition method. The surrogate, ENN, reduces query time to $O(N)$, and the acquisition method does not require uncertainty calibration.

## 3 Related work

There are many approaches to scale BO to many observations.

**Blackbox Matrix-Matrix Multiplication** A conjugate-gradient algorithm replaces the inversion of $K(x_m, x_m)$ in equation equation 1 with a sequence of matrix multiplies, reducing the query time complexity of a GP from $O(N^3)$ to $O(N^2)$ Gardner et al. (2018). A Lanczos algorithm can speed up GP posterior sampling (e.g., used in Thompson sampling) to constant-in-$N$ Pleiss et al. (2018).

**Trust Region BO** The TuRBO algorithm Eriksson et al. (2019) reduces wall-clock time in two ways: (i) It occasionally restarts, discarding all previous observations, resetting $N$ to 0. (ii) It Thompson samples only within a trust region, a small subset of the overall design space where good designs are most likely, thus avoiding needless evaluations elsewhere.

**Modeling $p_*(x)$** An open-source optimizer, Optuna Akiba et al. (2019); Optuna (2025), does not model $f(x)$. Instead, it models $p_*(x) = P\{x = \arg\max_x f(x)\}$. The model is a Parzen estimator, a linear combination of functions of the observations, which has $O(N)$ query time. Optuna uses a modified EI-based acquisition method Watanabe (2023).

Another optimizer, CMA-ES Hansen (2023), an evolution strategy (not a Bayesian optimizer), uses no surrogate at all. It models the distribution of the maximizer, $p_*(x)$, by evaluating $f(x)$ directly in batches of designs. After each batch is evaluated, all previous observations are discarded. Thus, the compute time of a CMA-ES proposal is $O(1)$, constant in $N$.

Other methods of scaling to large $N$ replace the GP with a neural network Snoek et al. (2015) or a random forest Hutter et al. (2011). While fitting a neural network or random forest scales as $O(N)$, the fitting processes are complex and introduce many tunable hyperparameters. Query time depends on the model architecture and is independent of $N$.

Inducing point methods Titsias (2009) introduce $M$ summary points, reducing the training complexity of GPs to $O(NM^2)$ and storage to $O(NM)$, with prediction costs of $O(M)$ for the mean and $O(M^2)$ for the variance, independent of $N$. Optimally, $M$ increases only slowly with $N$ Burt et al. (2020). Stochastic variational GPs (SVI-GP) further reduce the per-minibatch complexity to $O(M^3)$, independent of $N$ Hensman et al. (2013). Fitting requires choosing inducing points (various approaches exist Moss et al. (2023)) and can be computationally intensive, since the variational loss is more complex than the exact GP likelihood. These methods become relatively efficient when $N \geq 10,000$ Wang et al. (2019).

Another approach to scaling is to model $f(x)$ using nearest neighbor observations, as in Vecchia GP methods Jimenez & Katzfuss (2022) and others Gramacy & Apley (2015); Wu et al. (2024). One Vecchia method explored in Jimenez & Katzfuss (2022) combines nearest-neighbor lookups with TuRBO's trust region sampling. This approach is most similar to ours.

In contrast to the approaches above, our method, TuRBO-ENN, drastically simplifies the surrogate model and dispenses with hyperparameter *fitting* altogether. (It has a hyperparameter; It just does not fit it. See Section 4.) As a result, our method scales linearly, $O(N)$, and has a smaller constant factor, making it faster at every tested $N$. See Figure 1.

In preparing Figure 1, we found that the Vecchia GP method Jimenez (2025) to take 3 - 4 times longer to run than TuRBO and, thus, have omitted it from further numerical studies.

Chen & Lam (2025) recently introduced *Pseudo-Bayesian Optimization (PBO)*, which establishes convergence guarantees for any method whose surrogate, uncertainty quantifier, and acquisition satisfy certain conditions. Appendix C shows that TuRBO-ENN meets these conditions and, therefore, qualifies as a PBO method, inheriting the associated convergence guarantees.

## 4 EPISTEMIC NEAREST NEIGHBORS

### 4.1 ENN SURROGATE

We define ENN by three properties. For a query point, $x$,

- **Independence**: Each observation, $(x_m, y_m) \in \mathcal{D}$, produces an independent estimate of $f(x)$.
- **Mean**: $\mu(x \mid x_m, y_m) = y_m$.
- **Epistemic variance**: $\sigma^2(x \mid x_m, y_m) = d^2(x, x_m)$, where $d(x, x_m)$ denotes the (Euclidean) distance from $x$ to $x_m$.

Precisely speaking, we *treat* the estimates as independent for tractability. Equating epistemic variance to squared distance from the measurement, $x_m$, captures the intuition that similar designs will have similar evaluations, $f(x)$.

**Combining estimates.** For a query point, $x$, we combine the estimates from its $K$ nearest neighbors into the linear combination with minimum variance, the precision-weighted average Cochran (1954)

$$\mu(x) = \frac{\sum_i^K \sigma^{-2}(x \mid x_i)\mu(x \mid x_i, y_i)}{\sum_i^K \sigma^{-2}(x \mid x_i)}, \qquad \sigma^2(x) = \mathrm{Var}[\mu(x)] = \frac{1}{\sum_i^K \sigma^{-2}(x \mid x_i)}$$

Substituting from the properties above yields the ENN estimator:

$$\mu(x) = \frac{\sum_i^K d^{-2}(x, x_i)y_i}{\sum_i^K d^{-2}(x, x_i)}, \qquad \sigma^2(x) = \frac{1}{\sum_i^K d^{-2}(x, x_i)} \tag{2}$$

**Computational cost.** Finding the $K$ nearest neighbors requires evaluating $d(x, x_i)$ for all $N$ observations, costing $O(N \ln K)$ time per query. Treating the observations as independent relieves us from calculating the $O(N^2)$ pairwise covariances between observations as in the calculation of $K(x_m, x)$ in equations 1. Our implementation uses the Python module Faiss Meta (2024b) to find the $K$ nearest neighbors.

**Hyperparameters** Note that we fix $K$ (ENN's hyperparameter) to a single value for the entirety of all BO runs. This contrasts with most other surrogate methods, which re-fit either surrogate hyperparameters (e.g., a GP may fit lengthscales, output scales, and other hyperparameters Gardner et al. (2018)) or model parameters (e.g., tree-based methods or neural networks fit their parameters to the observations). In this respect, our method is perhaps most similar to a Tree-structured Parzen Estimator (TPE) Watanabe (2023); Akiba et al. (2019): a TPE updates summary statistics of the

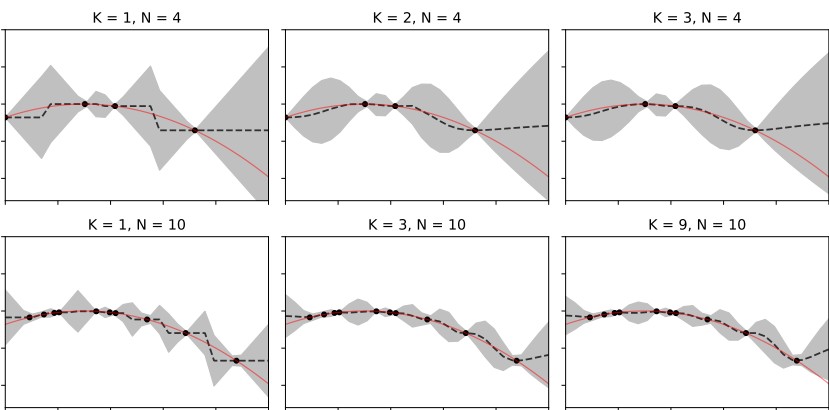

Figure 2: Epistemic nearest neighbors (ENN) surrogate. The dashed line shows $\mu(x)$ and the shaded region is proportional to $\pm\sigma(x)$. Only the relative size of $\sigma(x)$ is meaningful because $\sigma(x)$ is uncalibrated. The solid red line is the function being estimated, $f(x)$.

observations but does not perform an iterative fitting procedure to tune hyperparameters or parameters. The impact of the choice of K on performance is discussed in Appendix A.

Figure 2 depicts the ENN surrogate for an inverted parabola with various numbers of observations, $N$, and settings of $K$. Both the mean and uncertainty become smoother as $K$ increases. Notice in the lower right subfigure ($K = 9$, $N = 10$) that the red line falls outside the gray area. A fitted model would contain the function in $\mu(x) - 2\sigma(x) < f(x) < \mu(x) + 2\sigma(x)$, with high probability. In the next section, Section 4.2, we provide numerical evidence that surrogate fitting is not necessary for effective BO.

## 4.2 Acquisition via Non-dominated Sort

Because the ENN surrogate is not fit, the scales of $\mu(x)$ and $\sigma(x)$ are unrelated, and acquisition methods that compare $\mu(x)$ to $\sigma(x)$ (e.g., UCB, EI, TS) would yield arbitrary proposals. For example, $UCB(x) = \mu(x) + \beta\sigma(x)$. Unless $\sigma$ is properly scaled to capture the uncertainty in $\mu(x)$, $UCB(x)$ is meaningless. We, therefore, treat acquisition as a bi-objective optimization that maximizes both $\mu(x)$ and $\sigma(x)$, an approach that is insensitive to the overall scales of $\mu(x)$ and $\sigma(x)$ (and was previously studied in De Ath et al. (2021)).

**Pareto dominance.** Let $\mathcal{X} \subseteq [0, 1]^D$ be the search space. For any two points $x_i, x_j \in \mathcal{X}$, $x_i$ *dominates* $x_j$ if $\mu(x_i) \geq \mu(x_j)$ and $\sigma(x_i) \geq \sigma(x_j)$, with at least one inequality strict. The *Pareto front* $\mathcal{PF}(\mu, \sigma)$ is the set of non-dominated points. More on this may be found in Appendix B.

**Practical approximation.** We sample a finite candidate set $C = \{x_i\}$ uniformly inside TuRBO's trust region, compute $\mu(x)$ and $\sigma(x)$ for each $x_i$, and extract the non-dominated subset $\mathcal{PF}_0 \subset C$ via non-dominated sort Buzdalov & Shalyto (2014). A proposal, or *arm*, $x_a$ is drawn uniformly at random from $\mathcal{PF}_0$. If more than one arm is required, we keep sampling without replacement from $\mathcal{PF}_0$. If $\mathcal{PF}_0$ is exhausted, we find the next non-dominated subset, $\mathcal{PF}_1 \subset C \setminus \mathcal{PF}_0$. The process repeats until we have sampled the required number of arms. Code is available at [anonymized].

## 5 Numerical experiments

We benchmark TuRBO-ENN on three categories of problem: (i) 51 analytic test functions Surjanovic & Bingham (2013), (ii) Five reinforcement-learning environments (LunarLander-v3, Swimmer-v5, Hopper-v5, Ant-v5, Humanoid-v5, Farama (2024)), and (iii) The MOPTA08 automotive simulator Jones (2008). Across all tasks, TuRBO-ENN attains objective values comparable to TuRBO while reducing proposal time by an order of magnitude or more, with greater reductions for larger $N$. We compare to several baseline optimizers summarized in Table 1 and below.

The computations in this section required an estimated 30,000 cpu-hours of compute time.

Table 1: Optimization methods compared in Section 5.

| Method | Surrogate | Acquisition | Cost / proposal | RL Score | RL Time |
|---|---|---|---|---|---|
| random | none | Uniform sample | $O(1)$ | $0.082 \pm .018$ | 0.00023 |
| CMA–ES | none | Gaussian sample | $O(1)$ | — | — |
| optuna | Parzen KDE | Modified EI | $O(N)$ | — | – |
| lei | GP | Log-EI | $O(N^2)$ | — | — |
| ucb | GP | UCB | $O(N^2)$ | — | — |
| turbo-1 | GP | Thompson sample trust region | $O(N^2)$ | $0.35 \pm 0.017$ | 1.0 |
| turbo-0 | none | Uniform sample trust region | $O(1)$ | $0.25 \pm .004$ | 0.008 |
| turbo-enn-10 | ENN ($K = 10$) | Pareto($\mu,\sigma$) front | $O(N)$ | $0.32 \pm .005$ | 0.014 |

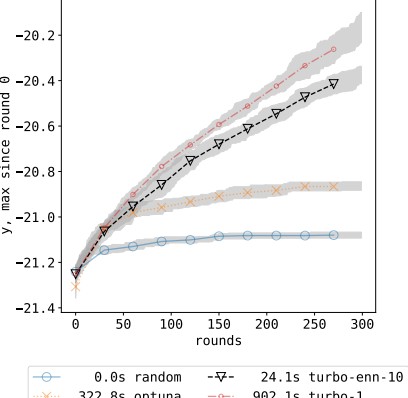

Figure 3: Four different optimizers optimizing a 300-dimensional Ackley function. The legend shows the total time (in seconds) spent calculating proposals.

## 5.1 SCORING

To introduce our comparison methodology, we compare TuRBO-ENN to other optimizers on the 300-dimensional Ackley function, Figure 3. In each round of the optimization, one design, $x_n$, is proposed and evaluated, $y_n = f(x_n)$. The max-so-far, $y_{\max,n} = \max\{y_0, \ldots, y(x_n)\}$, is plotted vs. $n$. We ran each optimization 10 times and depicted the mean of $y_{\max,n}$ by the dashed line and $\pm se$ (standard error) by the gray area.

We can summarize each optimization method's performance in Figure 3 with a single number, which we call the *score*. At each round, $n$, find the maximum measured values so far for each method, $m$: $y_{\max,n,m}$. Rank these values across $m$ and scale: $r_{n,m} = [\texttt{rank}(y_{\max,n,m}) - 1]/(M - 1)$, where $M$ is the number of methods. Repeat this for every round, $n$, then average over all $R$ rounds to get a score for each method: $s_m = \sum_n^R r_{n,m}/R$. The scores in figure 3 are $s_{\texttt{turbo-1}} = 1$, $s_{\texttt{turbo-enn-10}} = 2/3$, $s_{\texttt{optuna}} = 1/3$, and $s_{\texttt{random}} = 0$.

Using a normalized score enables us to average over runs on different functions which, in general, have different scales for $y$. Using a rank-based score prevents occasional, outlying result from dominating the average.

## 5.2 PURE FUNCTIONS

In these experiments we perform calculations similar to 3 for 51 test functions. To add variety to the function set and to avoid an artifact where an optimization method might coincidentally prefer to select initial points near a function's optimum (e.g., at the center of the parameter space Kudela (2023)), we randomly distort each function. We move the center point to a uniformly-randomly

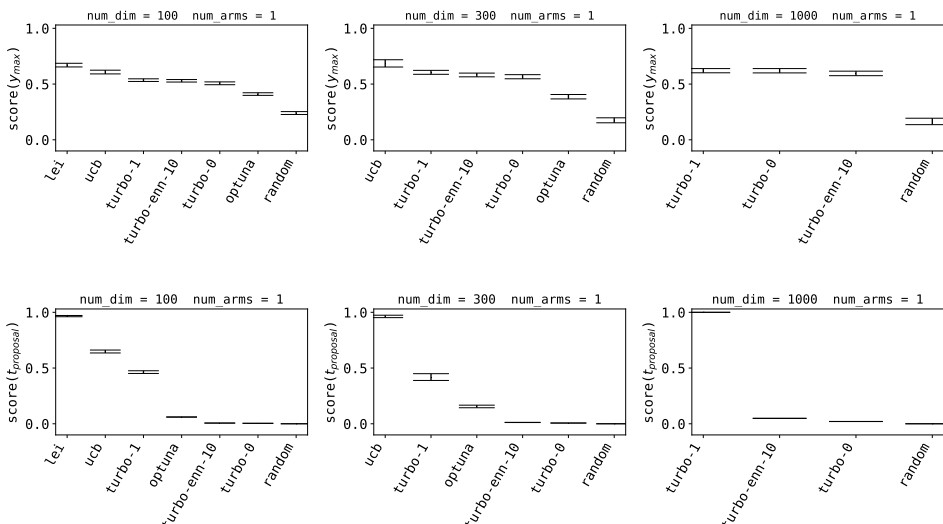

Figure 4: Optimizations on 51 pure functions in dimensions 100-1000. The top row compares the maximal attained objective values. The bottom row compares wall-clock time. TuRBO-ENN is faster than GP methods yet finds objective values comparable to those found by TuRBO-1.

chosen value, $x_0$. Along each axis, we distort like this

$$x' = \begin{cases} \frac{x - x_0}{1 + x_0} & x < x_0 \\ \frac{x - x_0}{1 - x_0} & x > x_0 \end{cases}$$

Note that the boundaries at 0 and 1 remain fixed. The value $x_0$ is set and fixed for the duration of an optimization run. We repeat the optimization 30 times for $D \leq 100$ or 10 times for $D > 100$, each time with a different random distortion. We optimize each function for $\max(30, D)$ rounds.

We calculate normalized scores for $y_{\max}$ as well as for $t_{proposal}$, the total time spent computing design proposals. Figure 4 compares TuRBO-ENN to TuRBO and other methods in various dimensions from 100-1000. Scores in each plot are averaged over 51 test functions. The full list of functions is available in the code repository [anonymized] and is taken from Surjanovic & Bingham (2013).

The other optimization methods are summarized in Table 1. The methods `lei` and `ucb` use a GP surrogate, and `optuna` uses a tree Parzen model. `turbo-1` is the single-trust-region variant of TuRBO Eriksson et al. (2019), which tracks a trust region (a box inside $[0, 1]^D$) from round-to-round. `turbo-1` Thompson samples from $\max(5000, 2D)$ candidate $x$ values, $C$, in the trust region. `turbo-enn-10` replaces the Thompson samples with samples from the Pareto front (see Section 4.2) of $C$ and uses ENN instead of a GP. We use $K = 10$ for ENN estimation. (Other values of $K$ are studied in Appendix A.) For all optimizations, the first round of arms (proposed designs) is chosen by some method of random initialization. For all TuRBO-based methods the initialization method is a latin hypercube design Santner et al. (2019). EI and UCB are initialized with Sobol' samples Santner et al. (2019).

We also include an ablation we call `turbo-0` in which we use no surrogate at all. Instead, we just select a design, $x$, uniformly randomly in the trust region. Comparing `turbo-0` to `random` and `turbo-1` allows one to disentangle the impact of the trust region logic from that of the surrogate. The column *RL Score* shows average scores (as defined in Section 5.1) on the three RL problems in Figures 5- 10. Sampling in the trust region instead of the entire bounding box improves the score from 0.21 (`random`) to 0.36 (`turbo-0`). Incorporating the ENN surrogate increases it to 0.42 (`turbo-enn-10`). Finally, switching from ENN to a GP further raises the score to 0.44 (`turbo-1`).

The column *RL Time* lists the total time a method spent calculating proposals, normalized to the time spent by `turbo-1`. The normalization was performed once for each problem since the proposal time varies by problem.

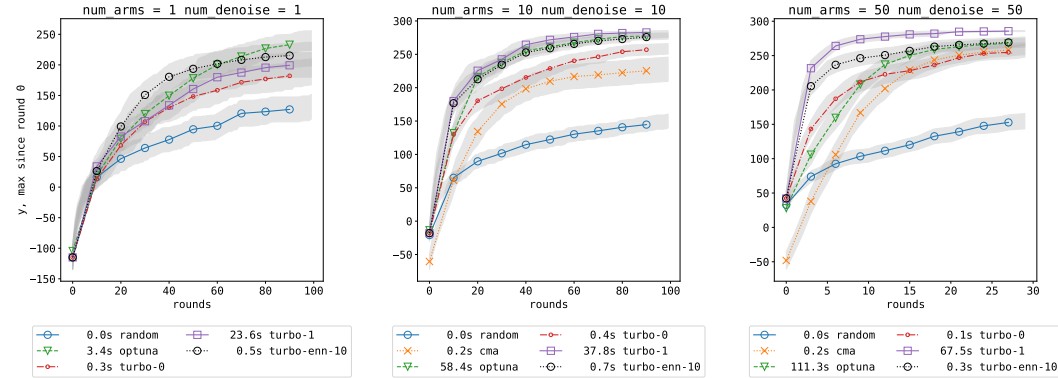

Figure 5: LunarLander-v3, $D = 12$, using the controller presented in Eriksson et al. (2019). `turbo-enn-1` performs comparably to `turbo-1`, which uses a GP, while achieving speeds nearly matching `turbo-0`, which uses no surrogate at all.

Figure 4 summarizes our results. We see that vanilla BO methods, `lei` or `ucb`, perform best (top row) but also take the most time to generate a proposal (bottom row). Note that when $D = 1000$, we exclude `lei` and `ucb` due to prohibitive computational demands. [1] `turbo-1` improves computation time but sacrifices some design quality. Finally, `turbo-enn-10` is significantly faster than `turbo-1` but still achieves comparable design quality.

The bottom row shows only the scores of each method's proposal time. For time in seconds, see figure 1. At the 1000th proposal, `turbo-enn-10` is over 30 times faster than `turbo-1`. Extending the curve for `lei` in the first plot to $N = 1000$, we estimate that `turbo-enn-10` would make its 1000th proposal over 1800 times faster than `lei`.

## 5.3 SIMULATORS

We next examine optimizations of several realistic problems. Most of the problems presented are from Gymnasium Farama (2024), the RL testbed originally known as OpenAI Gym Brockman et al. (2016).

For every gymnasium environment we use "frozen noise", i.e., each evaluation starts with a fixed random seed Kim et al. (2003). Panel (a) in each figure runs for 100 rounds using 1 arm per round. Panel (b), for each evaluation, averages the simulation's return (i.e., the sum of the rewards over all steps of the simulation) over `num_denoise = 10` different random seeds, proposing and evaluating 10 arms/round. Panel (c) averages over `num_denoise = 50` random seeds, proposing 50 arms/round. Configuration (c) was chosen to match the right panel of Figure 3 in Eriksson et al. (2019). All optimizations in this section were replicated 100 times, each time with a different set of random seeds, for variety. The legend shows the total time spent **proposing** arms over all rounds (averaged over replications), in seconds. The time required to run the **simulations** is excluded. Figures for other environments in Appendix D show the same patter.

In each case, we see `turbo-enn-10` producing designs of similar quality to those produced by `turbo-1`, but 10-100 times faster. Interestingly, the method `turbo-0`, which has no surrogate at all – it just samples uniformly from the trust region – outperforms other optimization methods on several problems. Hopper-v5 does not benefit from a surrogate, at least to within the error bars of our measurement. Nevertheless, including a surrogate (whether ENN or GP) generally further improves performance.

---

[1]Based on our runs of $D \in 30, 100, 300$, we project that a single optimization of a function in $D = 1000$ with `lei` would take nine hours.

## 6    LIMITATIONS AND FUTURE WORK

Improving scaling from $O(N^2)$ to $O(N)$ allows BO to handle many more observations. We would prefer, however, to see a constant-in-$N$, i.e. $O(1)$, scaling in a BO algorithm, as one finds in evolution strategies such as CMA-ES. In that case, the usable number of observations is unlimited.

Additionally, while TuRBO-ENN performs comparably to TuRBO (with a GP), neither perform as well (at least on the pure functions) as LEI or UCB. It would be interesting to seek a global ENN algorithm that matches the performance of LEI or UCB but retains the $O(N)$ scaling and speed of TuRBO-ENN.

In Appendix A, we discuss the potential to improve the dependence of ENN on $K$.

## 7    DISCUSSION

The major impediment to scaling to large numbers of observations in Bayesian optimization is the GP surrogate. Advances in GP methods have reduced the scaling from $O(N^3)$ to $O(N^2)$ for exact methods Gardner et al. (2018), but $O(N^2)$ still hampers scaling to BOMO problems. Scaling improves to $O(N)$ for approximate methods Titsias (2009); Moss et al. (2023) and alternative surrogates Snoek et al. (2015); Hutter et al. (2011), but fitting can be complex.

Our results show that BOMO problems can be solved to near state-of-the-art quality much more quickly by (i) using a simplified surrogate and (ii) omitting the surrogate-fitting step.

Almost certainly ENN provides worse estimates than a fitted GP. We do not optimize $K$, ENN has no other tunable parameters, and correlations between observations are ignored. But our goal in BO is not to estimate $f(x)$, it is to find its maximizer. TuRBO-ENN does this does this well and does it quickly.

We propose two explanations for this finding.

1. **Trust region** TuRBO's adaptive trust region biases samples to lie near the incumbent (best-so-far) observation. Table 1 shows that *even with no surrogate at all* (i.e., `turbo-0`), BO performance is significantly better than with random search (`random`). In fact, the bulk of the performance improvement between random search and the full TuRBO algorithm can be accounted for by the trust region. [N.B.: In addition to confining samples to a subspace, TuRBO samples only a subset of dimensions, a technique called RAASP Rashidi et al. (2024b); Eriksson et al. (2019). RAASP and the trust region, together, are responsible for the impact. Rashidi et al. (2024b) study the effects of RAASP and the trust region separately.] Additionally, the smaller the region over which a surrogate is required to predict, the less function variation it will need to model. Thus, a simpler surrogate (e.g., ENN) might incur less of a penalty in the overall BO problem when a trust region is used.

2. **Acquisition** In Bayesian optimization, the distribution of observations is not arbitrary: Observations are biased by epistemic uncertainty toward locations where the surrogate predicts with low confidence and toward locations of high $f(x)$. By these two mechanisms, observations tend to lie where they are needed for future acquisition decisions. We hypothesize that any surrogate that relies on nearby observations to predict at a point $x$, will fare relatively better in a BO setting that in an general regression problem. This may be interesting to make more precise in future work.

## 8    CONCLUSION

This paper asked whether we could speed up Bayesian optimization in the presence of many observations by removing the $O(N^2)$ bottleneck, the Gaussian process (GP) surrogate. Our numerical studies indicate that the answer is *Yes*. We introduced a simple model, Epistemic Nearest Neighbors (ENN), and found that substituting it for the GP in TuRBO resulted in one to two orders of magnitude reduced proposal time in our tests and better scaling ($O(N)$) without significantly sacrificing the quality of the proposed designs.

ACKNOWLEDGMENTS

We used ChatGPT and Cursor to help with writing code, constructing LaTeX commands, researching related work, and editing and revising this paper. All content remains our responsibility; we verified correctness and originality. *Otherwise anonymized until final submission.*

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

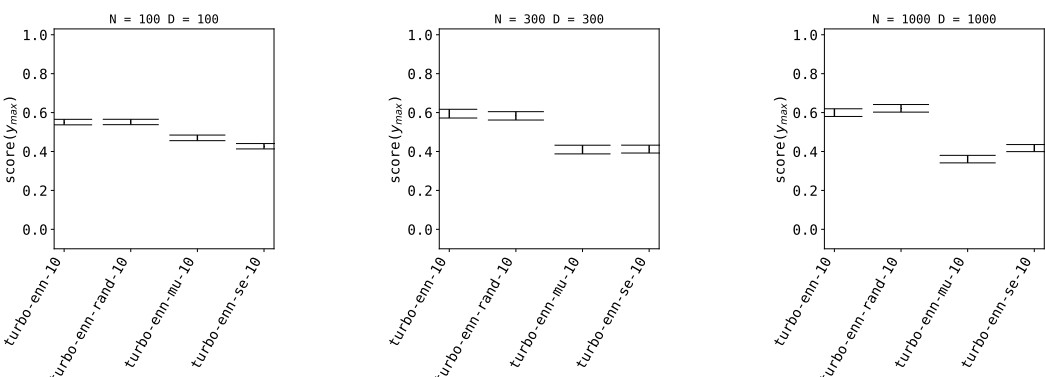

Figure 6: It is important to consider both $\mu(x)$ and $\sigma(x)$ in the acquisition method.

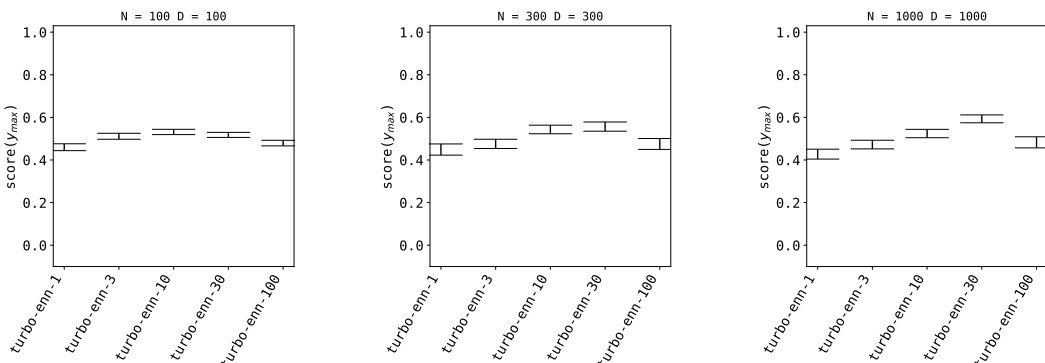

Figure 7: Score has a maximum in $K$.

## A    ABLATIONS

Our acquisition method (Section 4.2) relies on both the ENN mean $\mu(x)$ and its uncertainty $\sigma(x)$. To test whether each component is necessary, we evaluate three ablations of `turbo-enn-10`:

- **`turbo-enn-mu-10`**: propose the design with the largest $\mu(x)$ and ignore $\sigma(x)$.
- **`turbo-enn-sigma-10`**: propose the design with the largest $\sigma(x)$ and ignore $\mu(x)$.
- **`turbo-enn-rand-10`**: replace $\sigma(x)$ by a uniform random value $u \sim \mathcal{U}(0,1)$ when constructing the Pareto front, i.e. use $\mathcal{PF}\big(\mu(x), u\big)$.

The first two ablations simply ignore one of $\mu(x)$ and $\sigma(x)$. The third tests whether our model for $\sigma(x)$ carries any useful information. It could be that random numbers drive exploration just as effectively, in which case we could simply omit our $\sigma(x)$ estimates.

If both statistics contribute meaningfully, the full `turbo-enn-10` should outperform these ablations.

TuRBO-ENN comes with a hyperparameter, $K$, which determines the number of neighbors used to form estimates. Figure 7 explores the dependence of performance on $K$. The fact that there is a maximum in the function score vs. $K$ suggests that tuning $K$ at each round might be beneficial. Ideally, score would increase monotonically in $K$, which would enable one to subjectively trade off evaluation speed (low $K$) for design quality (high $K$). Future work could explore modifications of the ENN surrogate to achieve this goal.

## B    PARETO DOMINANCE

With two objectives the Pareto front is typically a one-dimensional curve (Figure 8); each point on the front dominates every point lying below or to the left of it but none on the curve itself.

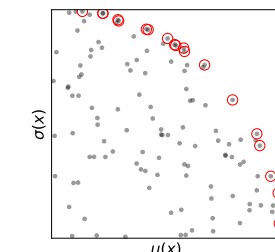

Figure 8: $\mu(x)$ and $\sigma(x)$ for 100 candidate design points. The circled (red) points are the non-dominated subset.

## C  TuRBO-ENN as Pseudo-Bayesian Optimization

Chen and Lam (2023) introduce the *Pseudo-Bayesian Optimization (PBO)* framework, which axiomatizes the conditions under which a black-box optimization method can be shown to converge. A PBO algorithm is defined by three components: a surrogate predictor (SP), an uncertainty quantifier (UQ), and an acquisition function (AF). For convergence, these must satisfy:

- **Local Consistency (LC):** The surrogate's prediction converges to the true function value in any region where data are dense.
- **Sequential No-Empty-Ball (SNEB):** Uncertainty is bounded away from zero in unsampled regions, and vanishes only when samples approach the point of interest.
- **Improvement Property (IP):** The acquisition assigns positive worthiness to any candidate with nonzero uncertainty, and converges to zero only when further improvement is impossible.

We now verify that TuRBO-ENN satisfies these conditions:

**LC**  TuRBO-ENN uses Epistemic Nearest Neighbors (ENN) as its surrogate, predicting the mean $\mu(x)$ from a weighted average of the $K$ nearest neighbors. Chen and Lam (2023) explicitly analyze $K$-nearest-neighbor regression and show that it is locally consistent for continuous $f$: with sufficient data near $x$, $\mu(x) \rightarrow f(x)$. Since ENN is a weighted $K$-NN model, it inherits this property and thus satisfies the local consistency condition.

**SNEB**  ENN defines the predictive variance $\sigma^2(x)$ using squared distances to the nearest neighbors. If $x$ is far from all sampled points, $\sigma(x)$ is large; as samples approach $x$, $\sigma(x) \rightarrow 0$. Hence ENN's uncertainty is SNEB-compliant: unexplored regions maintain positive uncertainty, and only regions with dense samples become certain.

**IP**  TuRBO-ENN selects new points via non-dominated sorting (NDS) on $(\mu(x), \sigma(x))$. Candidates with high $\sigma(x)$ but moderate $\mu(x)$ remain on the Pareto front, ensuring exploration whenever uncertainty exists. Conversely, if no improvement is possible with certainty, all candidates become dominated and their acquisition values vanish. This satisfies the improvement property.

Because TuRBO-ENN's SP, UQ, and AF satisfy the PBO axioms, it qualifies as a Pseudo-Bayesian Optimization method. Therefore, TuRBO-ENN is *algorithmically consistent* Chen & Lam (2025): its sequence of evaluations will eventually cover the search space, $[0, 1]^D$, and converge to the global optimum in the limit of infinite evaluations.

## D  More RL Problems

Figures 9 and 10 show `turbo-enn-10` producing high quality solutions in 10-100x less time than `turbo-1`.

Figure 11 shows results for three more problems, MOPTA08, Ant-v5, and Humanoid-b5. We excluded `turbo-1` and `optuna` from the comparisons of Ant-v5 and Humanoid-v5 because a single optimization would take longer than our allotted 5-hour window. We include them here simply

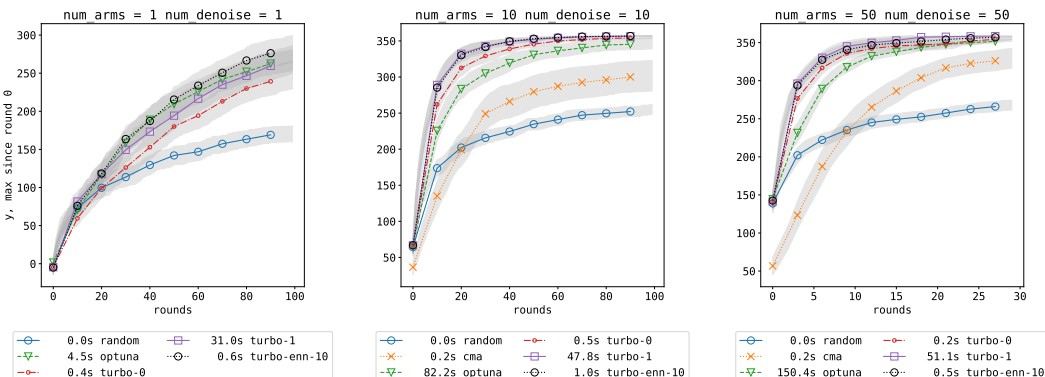

Figure 9: Swimmer-v5, $D = 17$, using a linear controller, similar to Mania et al. (2018). The controller designed by `turbo-enn-10` performs as well as that designed by `turbo-1`, but the designs are proposed almost 50 times faster.

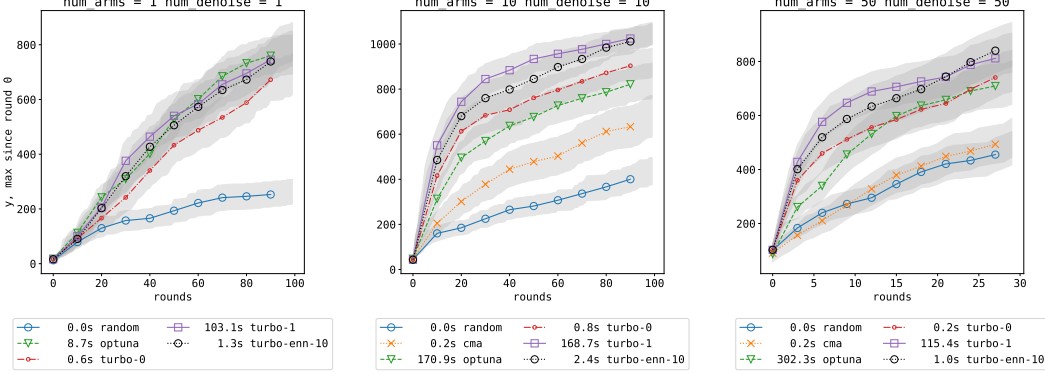

Figure 10: Hopper-v5, $D = 34$, using a linear controller, similar to Mania et al. (2018). Performance of `turbo-1`, `turbo-enn-10`, and `turbo-0` is comparable (within the error areas).

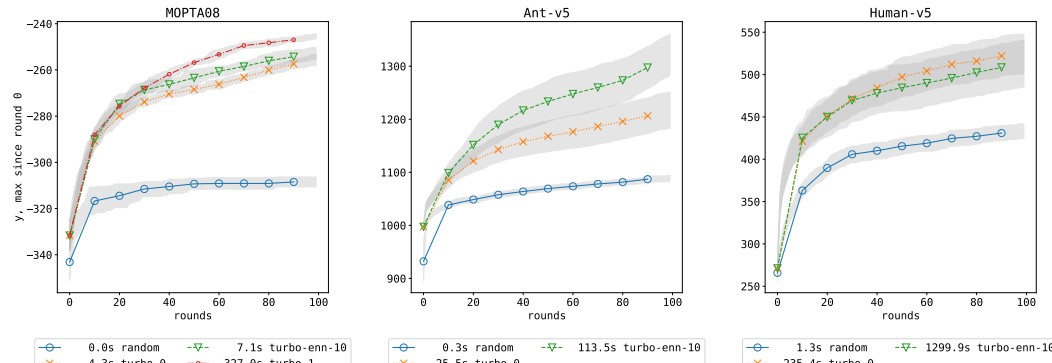

Figure 11: (a) MOPTA08 Jones (2008) $D = 124$ 10 arms/round (b) Ant-v5, $D = 841$, 100 arms/round. (c) Humanoid-v5, $D = 5861$. (b,c) use a linear controller, similar to Mania et al. (2018)

to demonstrate that it is possible to work with thousands of observations in hundreds or thousands of dimensions using our Bayesian optimization method.

