# OpenReview forum: "Taking the GP Out of the Loop"
_ICLR.cc/2026/Conference — Submitted to ICLR 2026_

### Official Review · Reviewer_bRuX · 2025-10-27

**Soundness:** 2
**Presentation:** 2
**Contribution:** 2
**Rating:** 2
**Confidence:** 5

**Summary:**

The paper proposes Epistemic Nearest Neighbors (ENN) as a lightweight alternative to Gaussian Processes (GPs) in Bayesian Optimization (BO), aiming to address scalability issues in BO with many observations. ENN estimates function values and an uncalibrated uncertainty using K-nearest neighbors, entirely avoiding GP hyperparameter fitting. To handle the lack of uncertainty calibration, the authors introduce a non-dominated sorting (NDS) acquisition strategy combining mean and uncertainty. The method, TuRBO-ENN, replaces the GP and Thompson sampling in TuRBO with ENN and NDS. Experiments on analytic benchmarks and RL simulators suggest comparable performance to TuRBO but with one to two orders of magnitude lower wall-clock time.

**Strengths:**

This paper identifies a genuine bottleneck in BO for large-N regimes and focuses on practical scalability rather than marginal gains in theoretical accuracy. ENN is simple: thus it maintains interpretability and offers immediate computational savings.

**Weaknesses:**

1. The core surrogate (weighted K-NN) is not new, while the acquisition strategy based on Pareto dominance between mean and uncertainty has been explored before (e.g., De Ath et al., 2021). Thus, the paper’s main contribution is engineering-oriented rather than conceptual.
2. There is no convergence or regret bound specific to ENN-based acquisition.
3. Because ENN’s “epistemic” variance is based purely on distance, it may not reflect model uncertainty in any principled sense. This undermines the interpretation of the “exploration” component in BO and raises questions about generalizability beyond trust-region settings.
4. The conclusion suggests “near state-of-the-art quality” while only showing parity with a reduced TuRBO baseline. The claim that the method “removes the GP bottleneck” is somewhat overstated given that GP approximations (e.g., inducing points, Vecchia) already achieve near-linear scaling.

**Questions:**

Please see Weaknesses.

---

> ### Author Response · Authors · 2025-11-14
>
> **The core surrogate (weighted K-NN) is not new, while the acquisition strategy based on Pareto dominance between mean and uncertainty has been explored before (e.g., De Ath et al., 2021). Thus, the paper’s main contribution is engineering-oriented rather than conceptual.**
> The addition of weighted epistemic uncertainty to weighted KNN is new, as is its use as a surrogate in BO.
>
> The application of Pareto dominance as a way to avoid costly calibration of uncertainty is also new. The insight that uncertainty calibration is not necessary for optimization is novel, and our results demonstrate the impact of that insight on computation time.
>
>
> **There is no convergence or regret bound specific to ENN-based acquisition.**
> Appendix C gives a convergence guarantee.
>
>
> **Because ENN’s “epistemic” variance is based purely on distance, it may not reflect model uncertainty in any principled sense. This undermines the interpretation of the “exploration” component in BO and raises questions about generalizability beyond trust-region settings.**
> The resolution of this dissonance is the novelty.
>
> Our notion of epistemic variance satisfies the improvement property (IP) of Pseudo-Bayesian optimization (lines 173-176 & Appendix C), which is one condition from which we guarantee convergence.
>
> The paper makes no claim about generalizing beyond trust-region setting, therefore any irrealis question raising would not relate to the fitness of the present work.
>
>
> **The conclusion suggests “near state-of-the-art quality” while only showing parity with a reduced TuRBO baseline. The claim that the method “removes the GP bottleneck” is somewhat overstated given that GP approximations (e.g., inducing points, Vecchia) already achieve near-linear scaling.**
> The reviewer's first claim is inaccurate. The work shows parity with TuRBO, the *exact reference code*. The phrase "reduced TuRBO baseline" has no connection to this work.
>
> In our studies, Vecchia GP's computational costs were so much higher than TuRBO's that it seemed pointless to include it. Our intention was to improve upon SOTA. SOTA is TuRBO, not Vecchia.
>
> Inducing point methods can take a very long time to fit. This is one bottleneck. While inducing point methods are ostensibly O(MN) for inference, popular reference code suggests taking M \propto N, which makes inference O(N^2).
>
> Any conception that Vecchia or inducing point methods are practical, scalable alternatives to TuRBO is not grounded in empirics.

---

> > ### Comment · Reviewer_bRuX · 2025-11-26
> >
> > I thank the authors for their clarifications. However, in my view, these clarifications do not change the quality or substance of the manuscript, and I therefore maintain my original score.

---

### Official Review · Reviewer_wmn2 · 2025-10-31

**Soundness:** 1
**Presentation:** 1
**Contribution:** 2
**Rating:** 2
**Confidence:** 4

**Summary:**

The paper proposes BOMO for Bayesian Optimization with Many-Observations. This is introduced in their TuRBO-ENN algorithm which replaces GP surrogate with a K-nearest-neighbors surrogate called Epistemic Nearest Neighbors (ENN) and avoids calibrated uncertainty by doing non-dominated sorting inside the TuRBO trust region.

**Strengths:**

The paper attempts to bring together several concepts to push Bayesian optimization (BO) to many observations settings. These, however, contradicts the very essence of BO which is typically targeted at optimizing black-box or expensive-to-evaluate functions with limited observations. The motivation presented by the authors read more like a succinct literature review with no clear case for such formulations.

Nonetheless, the authors attempted to think beyond conventional thinking by bringing together several formulations that are uncommon in BO settings to make a case for their proposition. Their formulations, however, lacks depth with their assumptions rife with several canonical settings including noise-free formulation, Euclidean-distance variance notion, and uncorrelated estimates of $f(x)$.

**Weaknesses:**

1. Unclear conceptual workflow; the proposition seems heuristically motivated with no clear theoretical formulations.
2. The empirical settings and results are unclear and lack comprehensive approaches. The authors claimed that another method (Vecchia GP) exists but seemingly provided a computational cost concerns as a basis for not comparing.
3. The performance of TuRBO-ENN as presented in the results are consistently behind conventional methods used for BO. The authors downplayed this as a separate focus despite making an earlier claim of making SOTA algorithm faster while producing comparable-quality solutions.
4. The paper is weakly presented with several mash up of literatures.

**Questions:**

1. The authors presented three properties which formed the basis of their work but mostly lacks justification. Can a sufficient narrative on the validity of those properties be presented?

2. The paper is rife with a lot of assumptions for simplicity including the properties in 1, noise-free derivation, Turbo-driven mechanics, and so on. A proper attempt to tackle the aforementioned problem in complex settings is needed. Can this be done?

3. At some points, the authors used the notion of D rather than N to make a case in their experiments. Can this be clarified as it contradicts the fundamental proposition of many-observations?

---

> ### Author Response · Authors · 2025-11-14
>
> **Unclear conceptual workflow; the proposition seems heuristically motivated with no clear theoretical formulations.**
> The workflow:
> - We seek to remove N^2 scaling bottlenecks.
> - This motivates remove GP surrogate.
> - Prior work mostly improves inference scaling; the fitting step still suffers.
> - This motivates removing the fitting step.
> - We propose ENN, which doesn't need fitting. ENN has uncalibrated uncertainty.
> - Uncalibrated uncertainty motivates an alternative acquisition method.
> - We propose non-dominated sorting (NDS) as the alternative acquisition method.
> - TuRBO-ENN satisfies the conditions that Pseudo-Bayesian optimization requires to guarantee convergence.
>
> **The empirical settings and results are unclear and lack comprehensive approaches. The authors claimed that another method (Vecchia GP) exists but seemingly provided a computational cost concerns as a basis for not comparing.**
> Our studies cover
> - 51 test functions over dimensions 100, 300, 1000
> - 5 RL environments, 3 of which are studied in three different batch & denoising settings, for effectively 15 different optimization problems, and
> - The MOPTA08 automotive simulator.
>
> It is rare in the literature to see so comprehensive a set of evaluations.
>
> In our studies Vecchia GP's computational cost was so much higher than TuRBO's that it seemed pointless to include it. Our intention was to improve upon SOTA. SOTA is TuRBO, not Vecchia.
>
> **The performance of TuRBO-ENN as presented in the results are consistently behind conventional methods used for BO. The authors downplayed this as a separate focus despite making an earlier claim of making SOTA algorithm faster while producing comparable-quality solutions.**
> This is not factual.
>
> In all test problems (sphere, ackley, etc.) TurBO-ENN's performance is within error bars of TuRBO (Fig. 4).
>
> In actual simulators, only one (Fig 5, right panel) of the 9 plots (Figs 5, 9, 10) is TuRBO clearly better than TuRBO-ENN.
>
> One point deserves clarification: TuRBO is SOTA among methods self-described as "scalable", which is the class of methods applicable to the simulation optimization problem on which this paper focuses.
>
> The EI and UCB methods are not scalable. We made certain to differentiate between "scalable in D" and "scalable in N". We will also differentiate between "scalable" and "not scalable".
>
> For large-scale simulation problems, the focus of this paper, EI and UCB are not viable options because they are not scalable in N.
>
> **The paper is weakly presented with several mash up of literatures.**
> I cannot parse "several mash-up of literatures". If the reviewer would care to clarify, I'd be happy to rebut.
>
>
> ## QUESTIONS
> **The authors presented three properties which formed the basis of their work but mostly lacks justification. Can a sufficient narrative on the validity of those properties be presented?**
> The discussion appears on lines 189-190, immediately following the presentation of the properties. To wit,
>
> Independence: Appears for tractability. This property makes it unnecessary to compute the ~N^2 covariances (see also lines 205-209).
>
> Epistemic variance: Similar designs, x, will have similar evaluations, f(x).
>
> The mean property is merely stated because we trivially form estimates of f(x) by each measured y.
>
>
> **The paper is rife with a lot of assumptions for simplicity including the properties in 1, noise-free derivation, Turbo-driven mechanics, and so on. A proper attempt to tackle the aforementioned problem in complex settings is needed. Can this be done?**
> Properties in 1: Addressed in previous question
>
> Noise-free: To communicate and justify the importance of noise-free simulations in science and engineering, we describe example applications and relevant technological trends, and we cite several sources of topic-focused research spanning the past 50 years. Additionally, we develop and/or compare to TuRBO, CMA-ES, and Optuna, all of which are in common use by engineers for optimization of noise-free simulations.
>
> (assume) TuRBO-driven mechanics: I don't know what this phrase means.
>
> "and so on": This is not a valid criticism.
>
> I cannot decipher, "tackle the aforementioned problem in complex settings". Please clarify.
>
> **At some points, the authors used the notion of D rather than N to make a case in their experiments. Can this be clarified as it contradicts the fundamental proposition of many-observations?**
> This comment is inaccurate and provides no evidence of its claim, while the paper provides copious evidence against.
>
> In the on boldfaced sentence (lines 50-51) in this paper we ask, "Can we make a SOTA algorithm significantly faster on BOMO problems while producing comparable-quality solutions?", which follows, "This work focuses on BOMO," making it clear that the paper is concerned with scaling in N.
>
> The paragraph before explicitly draws a distinction between high-D and high-N problems.

---

### Official Review · Reviewer_FdhK · 2025-11-01

**Soundness:** 3
**Presentation:** 3
**Contribution:** 2
**Rating:** 4
**Confidence:** 4

**Summary:**

This paper proposes a new surrogate model for Bayesian optimization.
The key motivation is that the commonly used surrogate model, GPs, has a cubic time complexity, which scales poorly when the number of queries increases.

To this end, this paper proposes epistemic nearest neighbors (ENN) as the BO surrogate.
This surrogate does not require model fitting.
The predictive mean is a weighted combination of the \\(K\\) nearest neighbors' labels, and the predictive variance is harmonic mean of the distances to the \\(K\\) nearest neighbors.

Empirically, the proposed surrogate (with a custom acquisition strategy and trust regions) yields competitive BO performance on many tasks.
Meanwhile, the surrogate model runs very fast as it does not require modeling fitting.

**Strengths:**

1. The proposed surrogate model is very simple.
It is based on \\(K\\) nearest neighbors, which could scale up to very large datasets thanks to highly optimized libraries nowadays.
The downside, however, is that the prediction of the ENN is often mis-calibrated.
And thus this surrogate model cannot be used with standard acquisition functions.
Instead, they have to resort to sampling non-dominated points in the trust regions.

1. It is also quite promising that even this simple model yields quite competitive BO performance.

**Weaknesses:**

1. The proposed method does cause BO performance regression, though the regression is often not significant.

1. The biggest concern that I have is that the experiments are not the best to validate the proposed method.
In particular, the maximum number of queries is 300.
At this scale, GP inference is still very fast despite the cubic time complexity.
Thus, it would be better to validate the proposed method in high throughout settings with tens of thousands of evaluations.
At the current stage, it is still not entirely clear if the proposed method is of practical use.

1. There is another potential concern when applying this surrogate model to high dimensional problems.
The proposed surrogate model is not compatible with existing acquisition functions because the predictive variance is often mis-calibrated (and also probably because the predictive mean/variance are non-smooth).
As a result, this surrogate model has to rely on discretizing the trust region and non-dominated sort on the discretization.
However, as the dimension increases, the size of the discretization should also increase, possibly exponentially.

**Questions:**

1. One thing would be interesting to try is running non-dominated sort acquisition for GPs.
By using the same acquisition, this checks how much the performance deterioration is caused by the surrogate model.

1. Since the proposed method also employs a trust region.
Does the trust region shrink/expand in the same way as TuRBO?
I am curious how often do the \\(K\\) nearest neighbors fall into the trust region.

---

> ### Author Response · Authors · 2025-11-14
>
> **The proposed method does cause BO performance regression, though the regression is often not significant.**
> This is overstated.
>
> In all test problems (sphere, ackley, etc.) TurBO-ENN's performance is within error bars of TuRBO (Fig. 4).
>
> In actual simulators, in only one (Fig 5, right panel) of the 9 plots (Figs 5, 9, 10) is TuRBO clearly better than TuRBO-ENN.
>
> **The biggest concern that I have is that the experiments are not the best to validate the proposed method. In particular, the maximum number of queries is 300. At this scale, GP inference is still very fast despite the cubic time complexity. Thus, it would be better to validate the proposed method in high throughout settings with tens of thousands of evaluations. At the current stage, it is still not entirely clear if the proposed method is of practical use.**
> This is inaccurate.
>
> The maximum number of queries is 10,000 (Figure 11, Ant & Humanoid).
>
> Figures 5, 9, and 10 query up to 1,500 times, matching the scale used in the original Turbo paper.
>
>
> **There is another potential concern when applying this surrogate model to high dimensional problems. The proposed surrogate model is not compatible with existing acquisition functions because the predictive variance is often mis-calibrated (and also probably because the predictive mean/variance are non-smooth). As a result, this surrogate model has to rely on discretizing the trust region and non-dominated sort on the discretization. However, as the dimension increases, the size of the discretization should also increase, possibly exponentially.**
> This concern is addressed by TuRBO (prior work). Its adaptive trust region mechanism finds a scale (for the trust region) that is small enough that acquisition is effective. It is quite elegant.
>
> ## QUESTIONS
> **One thing would be interesting to try is running non-dominated sort acquisition for GPs. By using the same acquisition, this checks how much the performance deterioration is caused by the surrogate model.**
> This would make a nice addition to the ablations. We will add it.
>
> **Since the proposed method also employs a trust region. Does the trust region shrink/expand in the same way as TuRBO? I am curious how often do the  nearest neighbors fall into the trust region.**
> Yes. We use the TuRBO algorithm (in fact, we use the TuRBO reference code, modified only to change the surrogate and acquisition method).
> This is interesting, but it seems like it would be a better fit for a study of trust region dynamics and the interaction between trust region size and characteristics of the surrogate.

---

### Official Review · Reviewer_dj4H · 2025-11-04

**Soundness:** 2
**Presentation:** 2
**Contribution:** 2
**Rating:** 4
**Confidence:** 4

**Summary:**

This paper proposes TuRBO-ENN, a scalable alternative to Gaussian-process (GP) based Bayesian optimization (BO) for problems with many observations or BOMO. Traditional BO suffers from $O(N^{3})$ scaling due to GP fitting and hyperparameter optimization, which becomes prohibitive when data are plentiful. The authors introduce Epistemic Nearest Neighbors (ENN), a lightweight surrogate that estimates mean function values and epistemic uncertainty from K nearest neighbors. So no training, no kernel inversion and no hyper. optimisation. The query and acquisition cost with no hyperparameter fitting is $O(N)$. Because ENN’s uncertainty is uncalibrated, standard acquisition rules like UCB or EI are invalid; instead, the paper proposes a non-dominated sorting based acquisition, jointly optimizing for large predicted value and uncertainty.

**Strengths:**

- Linear scaling (O(N) -  avoids the GP fitting bottleneck, enabling thousands of observations.

- No hyperparameter fitting - eliminates costly hyper. optimization and matrix inversion.

- Deterministic surrogate - simple, stable, and fast to evaluate using nearest neighbors.

- Calibration-free acquisition - Pareto non-dominated selection, again without tuning, so fast.

- Comparable performance - achieves nearGP solution quality with 10–100× faster proposal times across noise free benchmarks.

**Weaknesses:**

- An obvious weakness is that all the numerical experiments are conducted in the noise free setting, the authors justify this choice by mentioning that many simulation based optimisation porblems are noise free.
- Since ENN’s variance isn’t a probabilistic posterior, it cannot distinguish epistemic from aleatoric uncertainty, so handling noise properly would be non-trivial. The authors push this to future work but I think this is a big caveat and i am uncertain if this framework would stand in those settings.
- The ENN surrogate takes no account of correlations between inputs as no posterior covariance is learnt. I believe this seriously limits its extrapolation abilities.
- This method is not applicable in settings where either posterior modelling is critical (with full covariance) or where there is inherent noise. So the method is only applicable in a very limited setting.

It’s basically a lazy learner, like K-nearest-neighbors regression repurposed for Bayesian optimization.

Typos and other minor issues:

 - Line 67, spelling
- Citations all over the paper are not in the correct format, they should be in parentheses for readability.

**Questions:**

- How does the performance of TuRBO-ENN depend on the dimensionality
D, given that distance measures become less informative in high-dimensional spaces?
- How does TuRBO’s trust-region size interact with 𝐾 or the non-dominated sorting procedure in any systematic way?
- What are the theoretical or empirical limits of scalability.. at what 𝑁 or 𝐷 does the nearest-neighbor search itself become the new bottleneck?
- I am wondering if the performance improvements partly stem from TuRBO’s trust-region mechanism rather than the ENN surrogate itself?

---

> ### Author Response · Authors · 2025-11-14
>
> **An obvious weakness is that all the numerical experiments are conducted in the noise free setting, the authors justify this choice by mentioning that many simulation based optimisation porblems are noise free**
> This is arbitrary and non-factual.
>
> To communicate and justify the importance of noise-free simulations in science and engineering, we describe example applications and relevant technological trends, and we cite several sources of topic-focused research spanning the past 20 years, including a 20-year-old text which opens with the line, "In the past 15 to 20 years". These works have been collectively cited tens of thousands of times. Additionally, we develop and/or compare to TuRBO, CMA-ES, and Optuna, all of which are in common use by engineers for optimization of noise-free simulations
>
> **Since ENN’s variance isn’t a probabilistic posterior, it cannot distinguish epistemic from aleatoric uncertainty, so handling noise properly would be non-trivial. The authors push this to future work but I think this is a big caveat and i am uncertain if this framework would stand in those settings.**
> The reviewer has imagined a problem outside the scope of this paper and is unable to imagine its solution. That is not a valid criticism of the present work.
>
> **The ENN surrogate takes no account of correlations between inputs as no posterior covariance is learnt. I believe this seriously limits its extrapolation abilities.**
> Your consternation at that fact juxtaposed with the paper's excellent empirical results are why this paper is interesting.
>
> This is a paper about Bayesian optimization not a paper about prediction. We are making the point that a high-precision, generally-capable surrogate model is not necessary for achieving high-quality Bayesian optimization results. The reward for foregoing the exactness of a Gaussian process surrogate is a dramatic savings in proposal time. With this line of research we ask how *little* calculation we need to do to get BO results.
>
> We don't calculate covariances yet the results show incontrovertibly that we do not need to. That is the interesting result.
>
> **This method is not applicable in settings where either posterior modelling is critical (with full covariance) or where there is inherent noise. So the method is only applicable in a very limited setting.**
> This method is applicable in the noise-free simulation setting, which is obviously important and of broad interest (see response 1). Results in the paper empirically demonstrate its applicability on an automotive-industry simulation and several robot-controller simulations.
>
> ## QUESTIONS
> **How does the performance of TuRBO-ENN depend on the dimensionality D, given that distance measures become less informative in high-dimensional spaces?**
> See Figure 4. We show performance for dimensions 100, 300, and 1000. TuRBO-ENN performs on par (within errorbars) with original TuRBO.
>
> **How does TuRBO’s trust-region size interact with 𝐾 or the non-dominated sorting procedure in any systematic way?**
> The trust region varies, adaptively from round to round, which K is fixed. Earlier methods use univariate sorting, and authors do not report any systematic interaction between trust region size and sorting.
>
> **What are the theoretical or empirical limits of scalability.. at what 𝑁 or 𝐷 does the nearest-neighbor search itself become the new bottleneck?**
> In the paper we study problems up to N=10,000 (fig. 11) and D=5861 (fig. 11). [Note, however, that D scaling is not the subject of this paper]. We find TuRBO-ENN spends at most a few minutes, cumulatively, generating proposals.
>
> This question is best analyzed by comparing
>    P = proposal time / round
> to
>    E = evaluation time / round
> One can admit a subjective constant, k, that quantifies a user's relative sensitivity to proposal time, then ask when is
>     P >= k E
> P = c N, and E is constant, so the algorithm is bottlenecked at
>    N => (k/c) E
> This motivates research into O(1) [i.e., P constant in N] algorithms, as mentioned in section 6.
>
> **I am wondering if the performance improvements partly stem from TuRBO’s trust-region mechanism rather than the ENN surrogate itself?**
> As well you should, given the discussion of this topic on lines 368-374.
> The relative impact of the trust region varies from problem to problem. (i) Overall it is very important. (ii) The effects of the surrogate and trust region are additive.

---

### Meta-Review · Area_Chair_ZJfo · 2026-01-09

**Summary:**

The most significant concerns that were raised (note here I am listing concerns *raised* not yet whether they were correct or not) were:

- The small evaluation budgets considered across problems when the story is explicitly about scalability.
- The method is evaluated in the noiseless setting. I'll go ahead and lump the "aleatoric vs epistemic" comment here, since the likelihood noise is presumably the thing the reviewer is complaining is missing.

There were also some concerns I am outright discounting here:
- ENN's epistemic uncertainty being derived solely from distance -> True of GPs as commonly used in BO also.
- Somewhat vague comments about presentation.
- Comments on convergence. To be honest, I'm taken with neither the reviewer comment nor the resolution here. Appendix C does contain such a proof, which the reviewer may have missed, but also proofs of convergence that essentially distill down to "well we eventually sample everywhere" are also possibly not what the reviewer meant by their point (not that it should be necessary).
- Comparison to approximate GPs. SVGPs and Vecchia etc are great but at these smaller data scales, they're actually quite expensive because of the sheer amount of sequential work that has to be done.

**Reviewer Concerns:**

I do think the authors addressed most comments, although it is clear in some comments that the authors were perhaps overly frustrated with the reviewers while writing. Some of the authors comments did perhaps stray a little far into impoliteness.

**Reviewer Scores:**

One reviewer explicitly stated they didn't change they're score, but also none of their concerns are the main ones I am basing my decision on.

I'm going to assume that FdhK would likely have increased their score. The evaluation budgets were clearly larger than indicated. That said the actual details for these evaluation budgets are *slightly* unclear in the paper. The definition of a "round" appears in close proximity to the plot only in Figure 11a and 11b I think -- I might suggest explicitly changing the plot titles to arms_per_round instead of num_arms or similar, in case people aren't reading as carefully and miss 412-421 (I know that in the context of reviewing it's strange to have missed this...).

For dj4H, the noiseless setting comment is less well addressed, in the sense that the authors insistence that it is "arbitrary and non-factual"  seems to me to miss the point. The primary defense raised by the authors against the noiseless-only complaint is that many noiseless functions are important.

I think "noiseless functions are unimportant" is a strawman of the reviewer's concern. Rather, the point is that completely noiseless optimization, especially local optimization, is substantially, provably, and quantifiably easier (even in the black-box setting under common assumptions!). To that end, high dimensional BO methods methods like GIBO that explicitly benefit from noiseless observations become *vastly* more sample efficient relative to TuRBO. One of the main *reasons* that GPs are chosen as surrogates is to deal with some amount of noise.

To the point that the paper is going to answer its own question at the end, "this paper asked whether we could speed up Bayesian optimization.. by removing the ... Gaussian process (GP) surrogate" in the affirmative, it's a pretty clear limitation that is mentioned nowhere, and the solution is also simple: just acknowledge the limitation and compare to high dim BO methods that benefit greatly from getting to assume the function is noiseless (e.g., GIBO outperforms TuRBO on many of the RL gym baselines for precisely this reason).

Beyond that, if you are simply running TuRBO via the TuRBO implementation unmodified, TuRBO loses sample efficiency by not getting to "know about" the noiseless assumption your making, because it may mistakenly model some variations in the function as noise while running. Does TuRBO do better or worse if you pin the likelihood noise to a small constant? I'm not sure, but it seems like a relevant experiment, given that you are explicitly highlighting *knowing* the function is noiseless in your response to the reviewer.

---

### Decision · Program_Chairs · 2026-01-26

Reject